# Using a Modified Delphi Approach and Nominal Group Technique for Organisational Priority Setting of Evidence-Based Interventions That Advance Women in Healthcare Leadership

**DOI:** 10.3390/ijerph192215202

**Published:** 2022-11-17

**Authors:** Mariam Mousa, Helena J. Teede, Belinda Garth, Ingrid M. Winship, Luis Prado, Jacqueline A. Boyle

**Affiliations:** 1Monash Centre for Health Research and Implementation, School of Public Health and Preventive Medicine, Monash University, Melbourne, VIC 3168, Australia; 2Epworth Healthcare, Richmond, VIC 3121, Australia; 3Monash Partners Academic Health Science Centre, Melbourne, VIC 3168, Australia; 4Rural Health, Faculty of Medicine, Nursing and Health Sciences, Monash University, Melbourne, VIC 3168, Australia; 5Health Systems and Equity, Eastern Health Clinical School, Monash University, Melbourne, VIC 3168, Australia

**Keywords:** healthcare, priorities, implementation, methodology, nominal group technique, delphi, leadership, women

## Abstract

Background: Few studies address how to prioritise organisational interventions that advance women in leadership. We report on the relevance, feasibility and importance of evidence-based interventions for a large healthcare organisation. This study supports the first stage of implementation in a large National Health and Medical Research Council funded initiative seeking to advance women in healthcare leadership. Methods: An expert multi-disciplinary panel comprised of health professionals and leaders from a large healthcare network in Australia participated. The initial Delphi survey was administered online and results were presented in a Nominal Group Technique workshop. Here, the group made sense of the survey results, then evaluated findings against a framework on implementation criteria. Two further consensus surveys were conducted during the workshop. Results: Five priority areas were identified. These included: 1. A committed and supportive leadership team; 2. Improved governance structures; 3. Mentoring opportunities; 4. Leadership training and development; and 5. Flexibility in working. We describe the overall priority setting process in the context of our findings. Conclusions: With evidence and expert input, we established priorities for advancing women in healthcare leadership with a partnering healthcare organisation. This approach can be adapted in other settings, seeking to advance women in leadership.

## 1. Introduction

The under-representation of women in healthcare leadership has emerged as a critical concern for healthcare institutions, funding agencies and government [1,2,3,4]. Gender equity interventions to address this gender gap have been developed over the years, and are synthesised in a recent systematic review of organisational-level interventions that advance women in leadership [5]. Rationale for gender equity interventions range from a social justice position [6,7,8,9], where emphasis is on the provision of equal opportunities for men and women, to organisational performance as the context for change [10,11,12]. Organisational performance highlights the loss of talent and lack of diversity in approaches and decision-making, leading to a less than optimal capacity for improvement in healthcare outcomes [13,14,15,16,17]. In recent years, the responsibility of change has shifted from the individual to organisational action—on supporting women as a collective group, transforming strategy to recognise the deeply entrenched and seemingly intractable structural barriers and biases in policy, process and practice [5,18,19]. A greater emphasis has also been placed on building the knowledge base within the health sector, integrating what is known on gender-based research in the context of setting priorities for implementation within organisational practice [19].

Setting priorities for advancing women in healthcare leadership is highly relevant in the complexity of healthcare, the variation across health services and in the COVID-19 era of major demands and limited resources [20]. For gender equity interventions that advance women in healthcare leadership, current implementation challenges at the organisational level have included adhoc, inconsistent, often unstructured implementation of interventions that are often not evidence based or monitored for impact [5,15,19]. Formal priority setting processes can utilise best practice evidence, and engage the organisations leadership and workforce to identify what is most relevant and inform adoption of best practice interventions relevant to the organisational context in a constantly changing healthcare environment [21]. 

General methods for priority setting are well developed [22] yet often do not consider feasibility and implementation. To our knowledge, very few studies have shown how existing evidence is prioritised to meet organisational practice gaps in gender equity. In the context of a national funded partnership initiative to advance women, we aimed to apply robust, priority setting methods to assess the relevance, feasibility and importance of evidence-based interventions that advance women in healthcare leadership for a large private health organisation.

## 2. Methods

Using existing approaches to priority setting, our primary aim was to assess the relevance, feasibility and importance of evidence-based interventions that advance women in leadership, based on what was identified as gaps with gender equity practice in a large private healthcare organisation. The process is outlined in a priority setting program logic in Figure 1, which occurred in three phases: inputs, activities and outputs. In the first phase, an expert multi-disciplinary consensus panel comprising of health service professionals, including clinicians and clinical researchers was formed. This study was done in collaboration with one of Australia’s largest private healthcare networks following executive endorsement for panel participation. For context, Australia has a multi-payer public health system, supplemented by insurance and the private health system. A key input was to inform the development of the work using three data sources: (i) known evidence-based practice intervention categories extracted from a systematic review of the literature [5], (ii) findings from qualitative interviews, and (iii) by examining a local data source; the Victorian Gender Equality Action plan [23]. 

In the second phase, executive members of the expert group were pivotal in ensuring that there was an agreed approach to the work. A framework for implementation criteria (Framework S1: Organisational Priority Setting Framework) was developed and decided on through engagement with the organisation’s stakeholders. Expert group members participated in the initial Delphi survey (Survey S2: Delphi Survey). Results of this survey were used to develop a more focused list of interventions for testing against specific implementation criteria in the Nominal Group Technique (NGT) workshop, which included a sensemaking and a voting survey in order to reach consensus. Qualtrics, Poll Everywhere, and Zoom were software tools used to facilitate the priority setting activities. Qualtrics delivered the initial Delphi survey, Poll Everywhere was used during the workshop to facilitate voting and consensus for the NGT component and Zoom is where the workshop event took place. 

The culmination of findings from phase two of this study, systematically informed the development of a list of prioritised interventions that meet specific implementation criteria, set through engagement with the participating organisation’s stakeholders, and their overarching strategic plan [24]. This feeds into the development of an action plan as well as a communication plan to disseminate the findings more broadly for the organisation. Figure 1 provides a visual representation of how the overall Delphi and NGT contributed to the chain of outputs (phase 3), which directly inform the anticipated outcomes and their impact on advancing women in healthcare leadership. The program logic was part of the initial proposal, with ethics granted by the Monash University Human Research Ethics Committee (No. 25097). 

### 2.1. Delphi Process

Following design and planning, as well as expert panel selection (Figure 2), the next stage was to disseminate the initial Delphi survey (Appendix A) organisation wide to identify intervention gaps and priority needs. Dissemination was conducted by MM between May and June 2022 and supported by executive (LP) and research (IW) representatives at the participating organisation. One month prior to the planned NGT workshop, participants in the expert panel, and other participants who did not attend the NGT workshop, were sent a link to the survey via email, which contained a list of gender equity interventions for consideration. Interventions included in the survey were mainly extracted from three sources: (i) a recent systematic review of cross-sector evidence on organisational activities that advance women in leadership [5]; and (ii) interview data with women from the participating organization’s leadership who have experienced the career advancing impact of interventions (from within and outside of the organisation) [25]; and the Victorian Gender Equality Action plan [23]. Using a modified Delphi format, each participant was asked to select all interventions they believed most important for delivering gender equity outcomes that advance women into leadership at their organisation. Participants were also able to suggest additional priorities that were not listed. Mean ranking scores were computed for each priority, with higher scores indicating higher priority. Other questions were also included in the survey, related to implementation goals. Here, participants were asked about their beliefs around what is important for an organisation to do (future focused and action-based) in order to advance women in leadership, in addition to their current perceptions of the organisation, its capability, expertise, and possible resource and support requirements. The survey took on average 10 min to complete.

### 2.2. NGT Process

Intervention gaps and priorities identified in the round one Delphi survey, informed the content and purpose of the NGT workshop, which prioritised interventions based on expert panel consensus (Figure 2). The process was facilitated primarily by MM, with support from JB, HT and the broader Advancing Women in Healthcare Leadership (AWHL) project team. The NGT engaged experts in discussion to generate priorities, in a largely interpretive method [26,27]. With demonstrated validity, NGT is a process where all participants views can be equally considered, enabling consensus on the complex issue of gender equity that may otherwise invite askew decision-making by those who have a vested interest in a particular outcome [26,28]. Consensus methods such as the NGT are selected to overcome individual and group bias, through focusing the objective of the research, and developing priorities for action [26].

In this workshop, interventions were first discussed with the aim of making sense of what they mean for advancing women in healthcare leadership. This was followed by a confidential survey conducted live during the workshop for participants to privately rate interventions based on priority gaps. An adapted framework for judging implementation criteria (Appendix A) was explained within the context of the organisation; the same framework was used for group discussion and reaching consensus. The framework outlined 7 key feasibility areas for the participants to consider during the NGT workshop. These included the (1) potential for the proposed interventions to address current practice gaps; (2) the potential for the interventions to prevent significant adverse impact on women; (3) whether existing inequities can indeed be improved by the interventions across all regional locations; (4) whether there are clear drivers for stakeholders to engage and collaborate in implementing these interventions; (5) whether the problem or the potential solution align with current policy directions, including the organisations strategic priorities; (6) whether the interventions align with growth, and work to develop and succeed the workforce; and finally, (7) whether this can lead to a transformational change for women in healthcare and subsequently improve health outcomes. A final voting exercise was used to confirm the ranking of priorities and a list was formed for the organisation to consider.

## 3. Results

Both the Delphi and NGT participants in the priority setting program were comprised mainly of women (98%), who were medical practitioners (17%), nurses (14%), allied health professionals (21%), and hospital administrators (26%), occupying various levels of leadership and non-managerial positions (Table 1). Results of the Delphi and NGT are shown in Box 1 and Box 2, respectively.

Box 1Summary of the Initial Delphi survey findings.
**Delphi Survey 1 Initial Ranking**
Of all priorities listed in the survey, leadership commitment and accountability related interventions were seen as most important in ensuring gender equity remains a focus for the organisation. The top ten list of interventions included: committed and supporting leadership teamflexibility in work scheduleflexibility with work locationflexibility in work hoursleadership training and developmentsupportive and positive organisation cultureinformal mentorship for womenapproaching women directly with leadership opportunitiesformal mentoring program for womenprovision of scholarships for career development.Outside of the top 10 interventions, gender pay parity was a key topic added for discussion in the workshop, as was transparency in promotion processes, and the provision of child care services.

Box 2Summary of NGT findings.
**Survey 2—NGT Workshop Sense-Making, Group Discussion and Importance Voting**
The group discussion and sensemaking process collapsed a few of the listed intervention areas into one another if they were seen to be similar or to hold comparable strategic actions. A vote resulted in 4 clear areas of intervention seen as important priorities for the organisation. These included: (1) committed leadership and culture; (2) flexibility in work; (3) active leadership training and development—including sponsorship and scholarships and, (4) mentoring programs. A need for clear goals around each of these intervention areas were articulated, as an important part of good governance and transparent leadership commitment and culture. Potentially tangible metrics were discussed as part of ensuring that the importance of interventions and their associated actions was communicated, including the proportion of women in leadership roles within the organisation, the proportion of women currently under the guidance of mentors, and the proportion of training and development offers versus those actually accepted. Having these metrics built into a robust Information Technology (IT) infrastructure was seen as a necessary prerequisite. Flexibility was seen as multi-faceted, with special attention needed to support leaders who work out of office hour times, as well as those with child care responsibilities, with the provision of breastfeeding facilities. Flexibility here was defined as flexible approaches to work schedules, work locations and work hours. Furthermore, it was noted that the quality of mentorship needed to be managed and assessed, through the development of a formal matrix approach. Finally, there was consensus that all of the above needed to take an additional intersectional lens in order to get a clearer picture of the current state for all women in the organisation.
**Survey 3—NGT Workshop Group Discussion and Final Feasibility Voting**
After considering the new list of priorities against the feasibility framework, the final vote generated a total of 5 priorities. Support for child care was dropped as it was considered unfeasible to provide, considering the infrastructure requirements and a discussion on the importance of caring duties more broadly ensued. A vote amongst the group saw interventions that support caring duties more broadly (including caring for elderly parents, and caring for others with disabilities), was better suited under the umbrella of flexibility interventions. The gender pay gap, and transparency in promotion processes carried through as critical for progress and were placed under the basic governance intervention priority. Survey 3 feasibility rankings are presented in Table 2.

**Table 2 ijerph-19-15202-t002:** Descriptions of prioritised interventions based on feasibility of interventions, in order of importance.

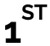	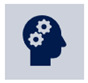	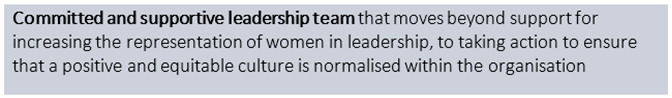
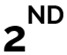	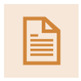	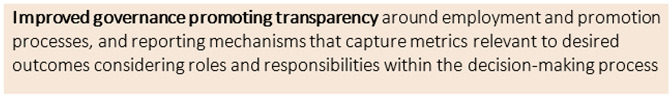
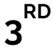	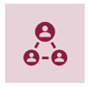	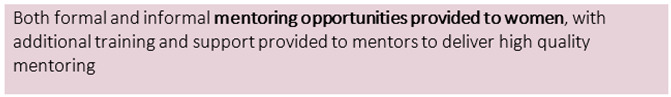
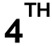	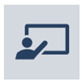	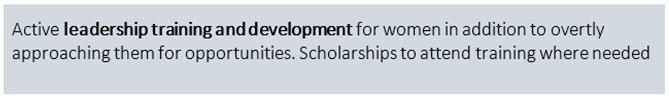
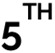	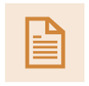	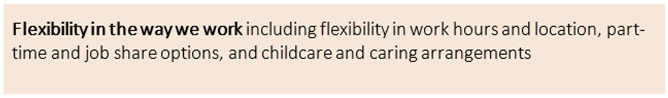

We based final priorities on outcomes of the NGT. Five priority areas were identified based on intervention importance and feasibility. These included interventions that supported: (1) a committed and supportive leadership team; (2) improved governance structures; (3) mentoring opportunities; (4) leadership training and development; and (5) flexibility in working. The prioritised interventions are described in more detail in Table 2 in order of their importance. 

## 4. Discussion

To our knowledge, no other studies have focused on setting priorities for organisational gender-equity interventions to advance women in leadership. While many organisations have traditionally focused on interventions targeted at the individual—where women are expected to be the change-makers alone—research indicates that change is more likely to occur as a result of addressing structural problems and workplace culture at the organisational level [15]. Our research adds to this by prioritising relevant and effective organisational solutions and strategies to progress women in leadership, as opposed to continuing the discussion on inequity and barriers to women’s career progression [2,3,5,29]. Here, effective organisational strategies were classified into five intervention priority groups (Table 2) and include: (1) a committed and supportive leadership team; (2) improved governance structures; (3) mentoring opportunities; (4) leadership training and development; and (5) flexibility in working. 

These findings converge with literature that shows that leadership commitment and accountability are fundamental in sanctioning organisational change for women in leadership [30,31,32,33,34]. To ensure consistency in practice, organisational leaders need to support and encourage improved workplace behaviors towards gender inequity, with impact on the overall working environment [35,36,37,38]. Research also shows that reporting on metrics related to interventions with actionable goals (e.g., quotas or targets), as they relate to supporting women in leadership, can be significantly more effective compared with reporting requirements in isolation [37,38]. 

Priorities centered on mentoring opportunities within the organisation, aligning with literature that shows that formal mentoring programs can have a positive effect on women’s career achievement and satisfaction [39,40,41,42]. Components to successful mentoring experiences include the provision of relevant and practical transfer of knowledge in a non-competitive environment; encouragement and endorsement; and guidance for developing a network of peers [40,41,43,44]. Similarly, for leadership training and development programs, research shows that they are important in fostering changes in attitudes and behaviors related to participation in leadership roles [45,46,47], while also strongly correlating with individual role engagement. As a priority, this not only benefits the organisation, but also demonstrates leadership and commitment to supporting women’s advancement to leadership [45,47,48,49]. Flexible work may be the more complex of all priority areas, as interventions in this area need to be implemented differently across early, mid and late career stages with support given to mitigating the impact of specific career inflection points (that is, career transitions that vary by position and discipline) [50]. Organisations that assist women to access strategies for flexibility at work, also benefit from addressing awareness of policies that may or may not already be in place, enabling women to utilise them effectively [51,52]. Similarly, where it may not be feasible to retrofit onsite facilities that support the provision of childcare services to help women with children, this can be considered in planning of new health services.

The overall findings in this study highlight that barriers for women, often stemming from organisational constraints and culture, can continue to perpetuate systemic inequities in the workforce, not related to individual capability. For example, constraints around the provision of support for caring responsibilities, and how they impact on women’s access to leadership opportunities. What this means is that many of the priorities discussed here, can work to harness workforce capability should the organisation ensure that the conditions that women continue to work in, are indeed designed for them and their life patterns. Attempts to do this within the participating organisation are informed by previous work [5,19,25], but also through incorporating both top-down commitment in the organisation, and bottom-up engagement and buy-in at the workforce level, which is in line with what the research says about structural change for enhanced gender equity outcomes [53]. 

Research on interventions that address gender equity issues for women in leadership expose a series of dilemmas faced by organisations, highlighting the disparities between gender-equity goals and their fit with organisational goals [19,54,55]. What this suggests is a persistent dual agenda, whereby gender equity goals can never be at the forefront, weakening and delegitimising efforts to facilitate change and close the gap between practice and evidence-based interventions. Addressing the distance between research and what is needed to fill a practice gap, (where what an organisation knows and what is does is aligned), forms the next step in implementation planning. In principle, this research was completed in collaboration and partnership with organisational stakeholders, with a willingness to address any potential misalignment and solve collective problems for both gender equity and the organisation. 

Further research is needed to continue the drive towards implementation and improving outcomes for women. What remains is a need for further understanding of how to implement organisational change in a real setting, and to provide much needed guidance for the integration of interventions into sustainable practices that mobilise change at scale. The organisation involved in this research will now work towards realising the priorities identified in this study into an implementation plan. This will facilitate the application of the findings in a way that suits the local context and will continue to be undertaken with academic partners. 

## 5. Strengths and Limitations

The co-creation of priorities between academics, clinicians and leaders of the organisation that will effect change is a major strength of this study. While the lead researchers (MM, JB and HT) were facilitators in the workshop, they were very aware of the importance of facilitating in a nondirective manner and have undertaken this work before. The collective group of stakeholders will enable future efforts to plan for effective implementation, supported by the research team. Data collection with the participating organisation and its workforce occurred at the tail end of the COVID-19 pandemic and therefore were not as heavily impacted by its demands. While the impact, if any, remains unknown, we anticipate it will have had some influence on individual perceptions of the importance and urgency of priorities, though not necessarily in a disadvantageous way for gender equity goals. A longitudinal view of organisational priorities beyond this time, would be of value. This study is being replicated across multiple organisations in the context of the broader NHMRC funded initiative, and the Advancing Women in Health Leadership project, to ensure that experiences of women who lead in various settings are equally considered. 

## 6. Conclusions

This study aimed to describe the modified methods and the framework used to set organisational priorities for implementation and advance women in leadership. Our findings suggest that the Delphi and NGT method were effective in extracting participant views, which significantly narrowed down and shifted intervention priorities towards addressing key gaps in practice within the organisation, taking a top-down and bottom-up view of what might be needed for implementation success. The modified Delphi, NGT and framework for implementation criteria contributed to setting priorities for interventions that advance women in leadership, based on identified practice gaps in the participating organisation. Five priority areas included: interventions that supported: (1) a committed and supportive leadership team; (2) basic governance structures; (3) mentoring opportunities; (4) leadership training and development; and (5) flexibility in working. The organisation is now able to work with these areas to develop a subsequent action plan. 

## Figures and Tables

**Figure 1 ijerph-19-15202-f001:**
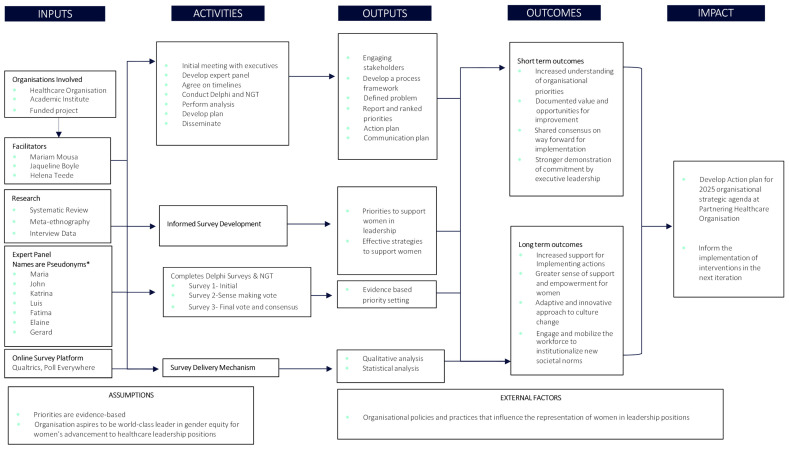
Priority Setting Program.

**Figure 2 ijerph-19-15202-f002:**
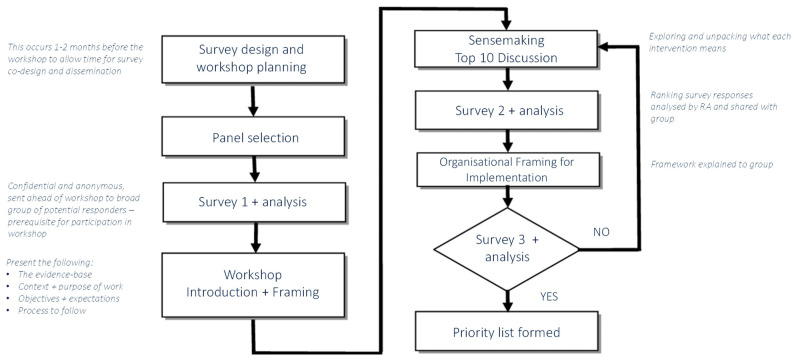
Workshop Process Map with embedded round 2 and 3 Delphi surveys and Nominal Group Technique.

**Table 1 ijerph-19-15202-t001:** Participant demographics in Delphi survey.

Position	
Executive level role	15%
Senior Management	23%
Middle Management	38%
Early career management	8%
Non-management position	18%
Role	
Medical Practitioner	17%
Nursing	14%
Allied Health	21%
Hospital Administration	26%
Research and Academia	5%

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
