# Peer review of "Using a Modified Delphi Approach and Nominal Group Technique for Organisational Priority Setting of Evidence-Based Interventions That Advance Women in Healthcare Leadership"

_ijerph, 2022, doi:10.3390/ijerph192215202_

Round 1

Reviewer 1 Report

Dear authors,

I would like to congratulate you for the effort that you have made for this paper. 

I appreciate the originality of your research and the interesting research design adopted as to better highlight the most important priorities which should be addressed to improve the participation of women in leadership.

The abstract is clearly written and provides the necessary information concerning the objective of the paper, the methods used as well as the results, but maybe it is a little bit longer than it should be. 

The manuscript is well structured and presented. Even so, I would like to suggest to the authors to pay more attention when inserting the numbers that correspond to the papers listed in the reference list. For instance, in the introduction section, we can see the associated numbers of the sources listed in the references section written with normal text, and in the final part of the section, written as superscript. 

The methods are well explained and reveal a hard work in separate sequences that complete each other. 

However, I would suggest to the authors to explain a little bit more the second priority - basic governance structures. Generally, this term governance structure is a specific notion for the New Institutional Economics and defines the institutional matrix within which a transaction is decided. According to Oliver Williamson, there are three types of governance structures: the market, the firm and the hybrid forms (referring to those that are placed somewhere in the middle, between the market and the firm). As far as I have understood, in the current study, when authors say governance structures, they are referring to specific rules or procedures that have the power to influence the decision making process concerning the access of women to healthcare leadership. In such context, maybe it would be useful to further describe the most appropriate governance operating model for such endeavor and, why not, to highlight if there are differences between private and public healthcare sectors or the same priorities can be applied to both sectors.

Another recommendation but maybe for further research is to also take into consideration the cultural background that circumscribes the governance structure and the incentives provided for women leadership in healthcare. Maybe the cultural dimensions of Geert Hofstede applied for the particular case of Australia could better explain why these five priorities identified in the current research have the power to generate a change in this respect.

Also, when dealing with the access of women to leadership in healthcare, another interesting dimension which can be also exploited is related to the specific matrix that defines the healthcare systems - Beveridge, Bismarck, out of pocket model or the national health insurance one. Which one is specific to Australia and how much openness does the healthcare system provide when dealing to women in leading positions?

The topic is very interesting and relevant considering the challenges brought by the covid pandemic, so congratulations for all your work!

Maybe some more attention can be paid to formatting part of the paper, but this is all!

Author Response

Reviewer 1 Comments:

Comment: The abstract is clearly written and provides the necessary information concerning the objective of the paper, the methods used as well as the results, but maybe it is a little bit longer than it should be. 

Response: Thank you for taking the time to review our manuscript and provide feedback. We have shortened the abstract, which now meets the journals guidelines of below 200-word count. Please see page 1 of the manuscript.

Comment: I would like to suggest to the authors to pay more attention when inserting the numbers that correspond to the papers listed in the reference list. For instance, in the introduction section, we can see the associated numbers of the sources listed in the references section written with normal text, and in the final part of the section, written as superscript. 

Response:  Thank you for pointing out this error. The referencing has been amended throughout to align with journal required formatting.

Comment: I would suggest to the authors to explain a little bit more the second priority - basic governance structures. Generally, this term governance structure is a specific notion for the New Institutional Economics and defines the institutional matrix within which a transaction is decided. According to Oliver Williamson, there are three types of governance structures: the market, the firm and the hybrid forms (referring to those that are placed somewhere in the middle, between the market and the firm). As far as I have understood, in the current study, when authors say governance structures, they are referring to specific rules or procedures that have the power to influence the decision-making process concerning the access of women to healthcare leadership. In such context, maybe it would be useful to further describe the most appropriate governance operating model for such endeavour and, why not, to highlight if there are differences between private and public healthcare sectors or the same priorities can be applied to both sectors.

Response: Governance structure in this instance refers to the framework of management specific to employment and promotion processes, and reporting mechanisms that capture metrics relevant to interventions that advance women in leadership. It also considers the roles of agents involved in these processes and their responsibilities within the decision-making process. This is now clarified in the manuscript (see Table 2, page 8). The goal of this paper is to primarily report on the findings of the priority setting exercise, within the context of the participating organisation. While we agree it would be of great interest to discuss the governance operating model, and the differences that may pose between private and public healthcare sectors, that would exceed the scope of this study. 

Comment:  Another recommendation but maybe for further research is to also take into consideration the cultural background that circumscribes the governance structure and the incentives provided for women leadership in healthcare. Maybe the cultural dimensions of Geert Hofstede applied for the particular case of Australia could better explain why these five priorities identified in the current research have the power to generate a change in this respect.

Response: Thank you for this recommendation. Cultural background is an important consideration and is noted for future research through our nationally funded partnership initiative on advancing women in health leadership (AWHL) (https://www.womeninhealthleadership.org/). The current study informs our efforts in the broader initiative to implement evidence-based interventions into practice, which is currently being replicated across multiple settings, with partners across public and private hospitals, Colleges and medical education/membership bodies.

Comment:   Also, when dealing with the access of women to leadership in healthcare, another interesting dimension which can be also exploited is related to the specific matrix that defines the healthcare systems - Beveridge, Bismarck, out of pocket model or the national health insurance one. Which one is specific to Australia and how much openness does the healthcare system provide when dealing to women in leading positions?

Response: Australia has a multi-payer public healthcare system, supplemented by an insurance supported, private healthcare system. This study was done in collaboration with support from one of Australia’s largest private healthcare networks. This detail has now been included in the manuscript (see page 2, para 4).

Comment: Maybe some more attention can be paid to formatting part of the paper, but this is all!

Response: Formatting has been revised. Thank you.

Reviewer 2 Report

This is an important topic and the approach used by the authors to prioritize interventions and guide implementation is novel and useful.

However, the manuscript could benefit from some additional elaboration about what is meant by a committed and supportive leadership team.

What specific steps can institutional leadership take to ensure that the workplace environment is supportive of women and truly recognizes and values their contributions?

Also, the authors note in the intro and discussion that many past efforts have focused on ‘fixing’ women. But isn’t that a primary goal of 2 of the priorities identified by the authors? (mentoring and leadership training).

I certainly support the concept of training and mentoring, but neither will accomplish anything unless the culture of the institution values and supports women. Thus, some discussion about how the priorities are tightly interconnected would be helpful to people who are thinking about how to implement these strategies.

Author Response

Comment: the manuscript could benefit from some additional elaboration about what is meant by a committed and supportive leadership team.

Response: Thank you for taking the time to review our manuscript and provide feedback. The meaning of a “committed and supportive leadership team” is outlined in Table 2 (page 8) and this has been elaborated on (see page 9, para 1).

Comment: What specific steps can institutional leadership take to ensure that the workplace environment is supportive of women and truly recognizes and values their contributions?

Response: This is a great question and one that we are currently working on in our national partnership initiative with health services, colleges and member organizations (AWHL, mentioned above). It is beyond the scope of this current manuscript, which was primarily to report on the findings of the priority setting exercise within the context of the participating organisation, but we have a forthcoming paper that does address this.

Comment: Also, the authors note in the intro and discussion that many past efforts have focused on ‘fixing’ women. But isn’t that a primary goal of 2 of the priorities identified by the authors? (mentoring and leadership training). I certainly support the concept of training and mentoring, but neither will accomplish anything unless the culture of the institution values and supports women. Thus, some discussion about how the priorities are tightly interconnected would be helpful to people who are thinking about how to implement these strategies.

Response: We have clarified our wording in the Introduction (see page 2, para 1) and Discussion (see page 8, para 2). In doing so, we have removed mention of ‘fixing’ women. What we intended to convey was that the burden of change was previously on the individual, and now this has changed to a focus on organisational action to support women as a collective group.
